physical chemistry/materials science

nanostructures, hierarchical floral morphology, Warberg capacitance, ferroelectric response, magneto-electric coupling

**Author for correspondence:**
Shahid Atiq
e-mail: satiq.cssp@pu.edu.pk

This article has been edited by the Royal Society of Chemistry, including the commissioning, peer review process and editorial aspects up to the point of acceptance.

# Ethylene glycol assisted three-dimensional floral evolution of BiFeO₃-based nanostructures with effective magneto-electric response

Syed Kumail Abbas[1], Ghulam M. Mustafa[1], Murtaza Saleem[2], Muhammad Sufyan[3], Saira Riaz[1], Shahzad Naseem[1] and Shahid Atiq[1]

[1]Centre of Excellence in Solid State Physics, University of the Punjab, Quaid-e-Azam Campus, Lahore-54590, Pakistan
[2]Department of Physics, School of Science and Engineering (SSE), Lahore University of Management Sciences (LUMS), Lahore, Pakistan
[3]School of Materials Science and Engineering, South China University of Technology, Guangzhou, China 510640

SA, 0000-0001-8012-9012

Controlled growth of nanostructures plays a vital role in tuning the physical and chemical properties of functional materials for advanced energy and memory storage devices. Herein, we synthesized hierarchical micro-sized flowers, built by the self-assembly of highly crystalline, two-dimensional nanoplates of Co- and Ni-doped BiFeO₃, using a simple ethylene glycol-mediated solvothermal method. Pure BiFeO₃ attained scattered one-dimensional nanorods-type morphology having diameter nearly 60 nm. Co-doping of Co and Ni at Fe-site in BiFeO₃ does not destabilize the morphology; rather it generates three-dimensional floral patterns of self-assembled nanoplates. Unsaturated polarization loops obtained for BiFeO₃ confirmed the leakage behaviour of these rhombohedrally distorted cubic perovskites. These loops were then used to determine the energy density of the BiFeO₃ perovskites. Enhanced ferromagnetic behaviour with high coercivity and remanence was observed for these nanoplates. A detailed discussion about the origin of ferromagnetic behaviour based on Goodenough–Kanamori's rule is also a part of this paper. Impedance spectroscopy revealed a true Warburg capacitive behaviour of the synthesized nanoplates. High magneto-electric (ME) coefficient of 27 mV cm⁻¹ Oe⁻¹ at a bias field of −0.2 Oe was observed which confirmed the existence of ME coupling in these nanoplates.

# 1. Introduction

Nanostructures of multifunctional materials have been a subject of interest because of their exciting physical and chemical properties, mainly thanks to their high surface to volume ratio. Tremendous work has been done in the recent past to control, for instance, the size, morphology and growth of functional materials in order to tune their magnetic, electric and optical properties. The motivation has further triggered, as many characteristics of advanced functional materials are highly influenced by their spatial geometry [1]. Among the vast category of such functional nanostructured materials, multiferroics have played a vital role in transformation of many scientific breakthroughs into technological innovations [2]. For instance, multiferroic materials have turned more efficient in current power demands of transistors leading to low power usage in electronic industries [3]. In addition, these multiferroics have a wide range of applications in spintronic devices like tunnel magnetoresistance (TMR), multistate memory devices and high sensitivity magnetic field sensors [4].

Controlling the shape and size of micro and nanostructured multiferroics can significantly affect the physico-chemical properties of the materials [5]. In this regard, $BiFeO_3$ (BFO) nanostructures have been extensively studied for their multifunctional properties. Some earlier studies have already shown wide tunability of physico-chemical properties by steering their size and shapes [6]. For instance, photocatalytic properties were enhanced by Joshi et al. by designing BFO nanocubes [7]. Recently, BFO nanoflakes have been used as an electrode material with an improved rate and cycling performance for Na-ion batteries [8].

BFO, being the only existing single-phase material possessing both ferroelectric and magnetic orders at room temperature, has attracted considerable interest when synthesized in the form of nanotubes, nanoplates (NPLs) and nanorods (NRs) [9–11]. A weak ferromagnetic order is observed in BFO thin films and nanoparticles which are credited to the size confinement effects. BFO nanoparticles with size less than 62 nm break the long-range spiral spin structure which enhances its magnetic attributes compared with bulk counterparts [12]. Several studies have been performed in order to enhance the multiferroic characteristics of BFO nanoparticles in recent years. Reddy et al. [13] for example, report enhanced size-dependent magnetic and ferroelectric orders at room temperature in BFO nanostructures. Furthermore, doping of transition metal ions in BFO at A and B sites have also shown strong variation in multiferroic properties [14]. In addition, BFO has been reported as contaminated with some secondary phases, due to which doping of transition metal ions becomes much more necessary. The choice of proper substituents not only helps to get phase purity but also to tune effectively the magnetic and ferroelectric response of BFO. This is the reason that Co and Ni have been chosen as dopants in this work. Depending upon the morphological impacts on the functional materials, it was of immense interest to synthesize the size- and shape-dependent BFO nanoparticles to study their electric and magnetic properties.

In this context, here, we report a series of Co and Ni co-doped BFO nanostructures synthesized using a solvothermal process to get controlled morphological attributes and to relate these with their multifunctional characteristics. Phase formation mechanism of BFO powders involve the dissolution of Bi and Fe precursors in the presence of a base (KOH/NaOH) to form the precipitates of their respective oxides and then treat them hydrothermally [15–17]. Systematic investigations regarding synthesis of BFO nanoparticles have been performed previously, but a detailed examination of morphological dependency on multifunctional properties still needed to be explored.

# 2. Experimental

In this work, ethylene glycol (EG)-mediated solvothermal method has been used for the synthesis of Co- and Ni-doped BFO nanoparticles. For this purpose, pure BFO and a series of $BiFe_{0.9}Co_{0.1-x}Ni_xO_3$ (x = 0.00, 0.03, 0.05, 0.07 and 0.1) samples were synthesized and were named as BFO, BFCO, BFCNO-3, BFCNO-5, BFCNO-7 and BFNO, respectively. Analytical grade precursors such as $Bi(NO_3)_3 \cdot 5H_2O$ (≥98%), $Fe(NO_3)_3 \cdot 9H_2O$ (99.99%), $Co(NO_3)_2 \cdot 6H_2O$ (≥98%) and $Ni(NO_3)_2 \cdot 6H_2O$ (≥98.5%) all taken from Merck, USA, were first dissolved in EG. The benefit of the use of EG is that it acts both as a solvent and a reducing agent for the successful synthesis of nanoparticles of different shapes and sizes without the use of nitric acid [18]. It has been reported that nitric acid is used for complete dissolution of Bi nitrate but it could result in coarser powder [19]. After thorough stirring at room temperature, 6 M KOH solution was added in the above solution dropwise until the desired pH was reached. This causes the co-precipitation of $Bi^{3+}$ and $Fe^{3+}$ ions which are now ready for the

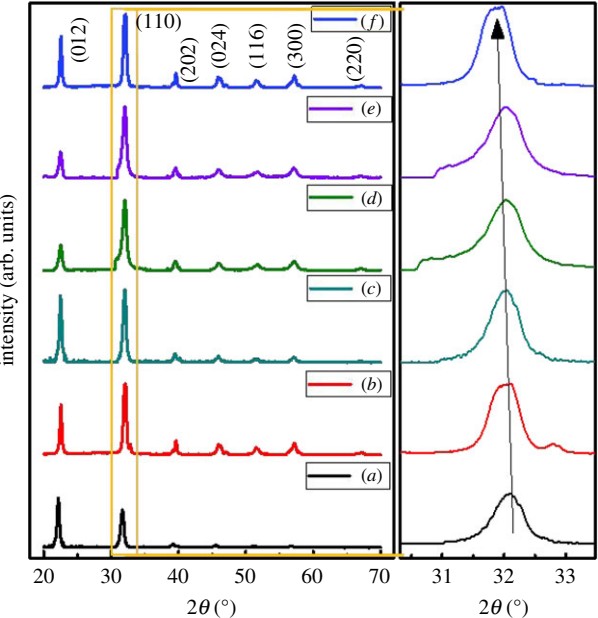

**Figure 1.** Indexed XRD patterns of the (a) pure BFO, (b) x = 0.00, (c) x = 0.03, (d) x = 0.05, (e) x = 0.07 and (f) x = 0.1 in $BiFe_{0.9}Co_{0.1-x}Ni_xO_3$.

hydrothermal treatment. This procedure was followed by the transfer of whole solution in a Teflon-lined autoclave which was then kept at 200°C for 6 h. After that, the autoclave was allowed to cool naturally and the sediments then obtained were washed several times with distilled water and dried at 60°C for several hours. The powder obtained was calcined at 600°C for 3 h in order to get the crystalline phase for the synthesized BFO ceramics.

Crystal structure for BFO samples was determined using a Bruker D8 Advance X-ray diffractometer (XRD) with a $2\theta$ range from 20° to 70° with a step size of 0.03°. XRD patterns were further analysed through Rietveld's refinement approach using X'Pert Highscore Plus software. To confirm the NPL synthesis, a NovaNano SEM-430, field emission scanning electron microscope (FESEM) was used at different magnifications starting from 5000x to 300 000x. FESEM was operated at a high voltage of 10 kV, with working distance of 5.2 mm and with through-lens detector (TLD). TLD is the most advanced detector used for detecting secondary electrons in order to generate high-magnification and high-resolution images. Ferroelectric and magneto-electric (ME) analysis was performed through Radiant Technologies Inc., USA, Precision Ferroelectric Tester (equipped with magneto-bundle and powered by Vision data acquisition software), with a current source, CS 2.5. An attached Helmholtz coil with a maximum of ±45 Oe field was used to generate uniform magnetic field. Tester was also used to obtain leakage current measurements and plotting of switching/unswitching graphs. Magnetization versus field (M–H) loops were obtained using a 7404-Lakeshore vibrating sample magnetometer (VSM). Frequency-dependent dielectric and impedance analysis was performed using 6500B, Wayne Kerr Precision Impedance Analyzer. For ferroelectric and dielectric measurements, the calcined powder was pressed into pellets through hydraulic presser having an area of 0.345 $cm^2$ and thickness 8 mm.

## 3. Results and discussion

Figure 1 shows the indexed XRD patterns of the synthesized nanostructures. The indexing method was adopted as mentioned by Cullity for rhombohedral structures [20]. All the indexed peaks were then matched with the ICSD ref. no. 01-071-2494 with $R3c$ space group which proved the formation of single phase of rhombohedrally distorted cubic structure. The indexed peaks led to the lattice parameter, $a$ in the range from 5.591 to 5.599 Å and $c$ from 13.855 to 13.863 Å, as shown in electronic supplementary material, table S1. This variation is confirmed from the inset of figure 1 in which a major peak is seen shifting slightly towards lower angle. The lattice parameters tabulated in electronic supplementary material, table S1 does not show a considerable change with the variation of Ni/Co ratio. It might be because of the comparable ionic radii of Ni (0.70 Å), Co (0.68 Å) and Fe (0.64 Å).

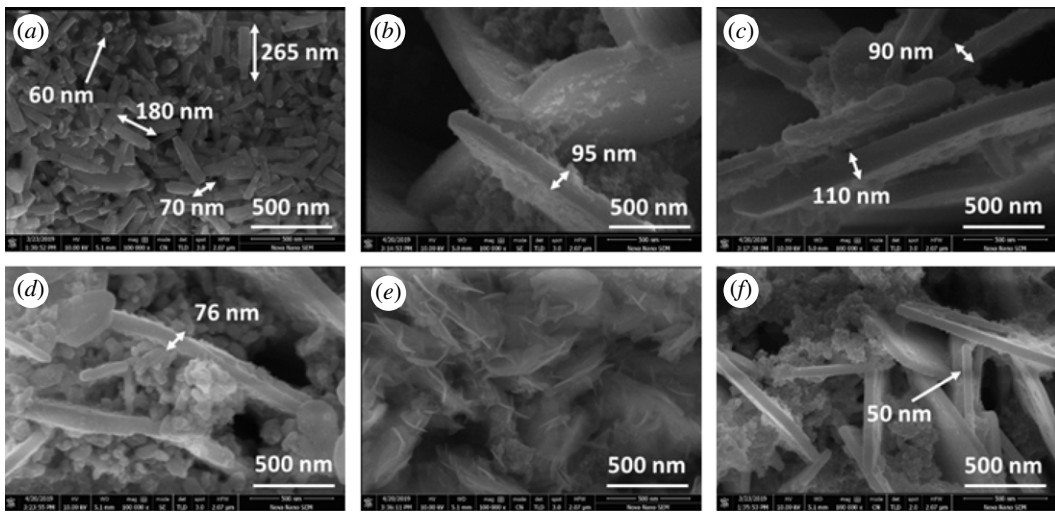

**Figure 2.** FESEM images of the (*a*) pure BFO NRs, (*b*) x = 0.00, (*c*) x = 0.03, (*d*) x = 0.05, (*e*) x = 0.07 and (*f*) x = 0.1 in $BiFe_{0.9}Co_{0.1-x}Ni_xO_3$ nanoplates.

The Rietveld's fitting method was further employed to study the variation in lattice parameters of BFO NPLs through X'Pert Highscore plus software. All the XRD patterns of parent and doped BFO NPLs were refined using the space group *R3c* and their refined patterns along with their difference plots are shown in electronic supplementary material, figure S1. No extra peak is seen in the XRD patterns and neither in their respective difference plots, which authenticated the phase purity of all the samples. This confirmed the formation of single-phase BFO NPLs occupying rhombohedral structure with *R3c* space group. The residual values or simply the *R*-values obtained during the refinement are a good way to judge the goodness of fit (GoF) [21]. Small *R*-values obtained from refinement confirmed that the XRD patterns were well fitted with the calculated patterns of *R3c* space group. Both the structural and residual parameters are tabulated in electronic supplementary material, table S1.

Figure 2 shows the FESEM images of pure BFO NRs and doped BFO NPLs at a high magnification of 300 000x. Highly magnified image of pure BFO showed uniformly scattered NRs with width in the range from 50 to 70 nm and length from 150 to 300 nm as shown in figure 2*a*. Most of the NRs were in uniform rectangular shape and their sizes were comparable to each other. Some NRs are placed horizontally while some other can be seen placed vertically so that only the tip of these NRs is visible. The size of the tip's width is about 60 nm which suggests a formation of 1D BFO NRs. Transformation of these NRs to NPLs was observed when BFO was doped with minute amount of Co and Ni. Figure 2*b–f* shows the NPLs formation for the Co- and Ni-doped BFO samples. In addition to these NPLs, some dispersed nanoparticles are also present. This type of growth mechanism has been reported as the anisotropic growth for the NPLs [22]. It is obvious from these images that some of the NPLs are curved and are interconnected or tangled within each other. The NPLs exhibit thickness values in the range of 50–110 nm, for different compositions of Co and Ni in BFO. Synthesizing nanostructures through a chemical method strongly depends upon the presence of $OH^-$ ions which directly affect the nucleation of oxide materials. $OH^-$ ions serve as a capping agent which creates anisotropic growth by adsorbing on certain faces of the oxide crystals [23]. This type of controlled growth has been adopted many times to synthesize one- and two-dimensional nanostructures [24,25]. In addition to NRs and NPLs, the composition BFCNO-7 shows very thin plates of really small width which could be regarded as nanoflakes.

Hierarchical flowerlike morphology from the arrangement of two-dimensional NPLs can be seen in figure 3. These NPLs were arranged symmetrically and were tangled in a way to form the self-assembled floral morphology. The dense core of these flowers indicates close packing of BFO NPLs at the flower centre. A variety in size of these hierarchical flowers can also be seen from figure 3*a*. Further insight into the flower can be seen from figure 3*b–d*. This type of morphology provides strong mechanical strength and mass transportation at the surface of these materials [26]. Electronic supplementary material, figure S2 gives the summary of the transformation of pure BFO NRs to doped BFO NPLs and finally the floral pattern made up of these self-assembled NPLs.

Ferroelectric behaviour of the material can be directly indicated by plotting the polarization versus electric field (*P–E*) loops. Figure 4*a* shows the bipolar polarization hysteresis loops of pure and doped BFO at RT from which remanence polarization ($P_r$), maximum polarization ($P_m$) and coercive field ($E_c$)

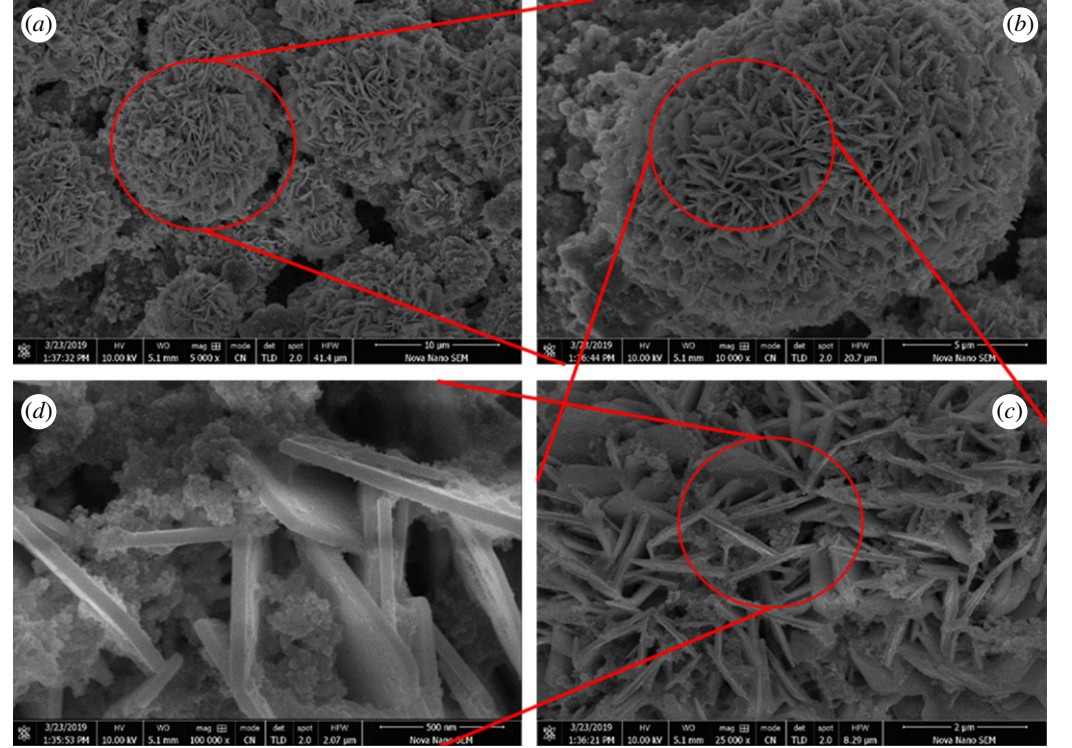

**Figure 3.** Insight view of the floral patterns of the self-assembled NPLs at different magnifications of (*a*) 5000x, (*b*) 10 000x, (*c*) 25 000x and (*d*) 50 0000x.

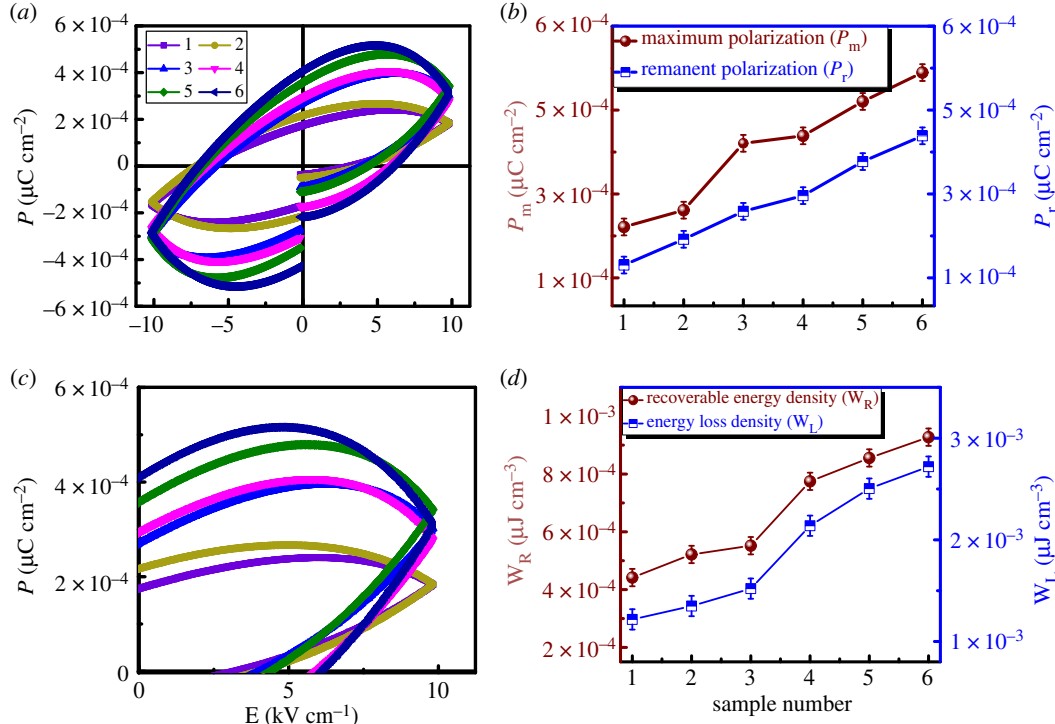

**Figure 4.** (*a*) *P-E* hysteresis loops, (*b*) $P_m$ and $P_r$ variation, (*c*) mono-polar *P-E* loops and (*d*) $W_R$ and $W_L$ plots of the (1) pure BFO, (2) x = 0.00, (3) x = 0.03, (4) x = 0.05, (5) x = 0.07 and (6) x = 0.1 in $BiFe_{0.9}Co_{0.1-x}Ni_xO_3$.

were obtained. *P–E* loops of pure and doped BFO were obtained at applied field of 10 kV cm$^{-1}$, indicating high resistivity and breakdown at higher applied electric fields [27]. It has been concluded that high voltage breakdown is the result of leakage current due to high conductivity of the BFO

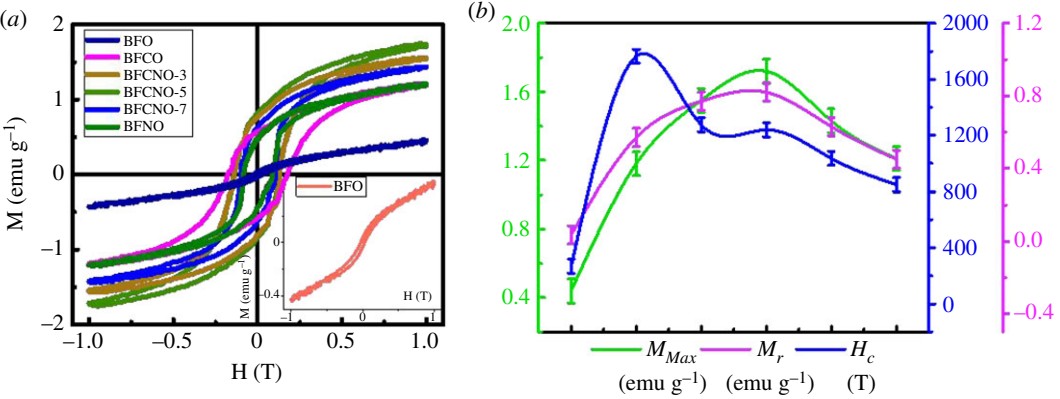

**Figure 5.** (a) *M–H* hysteresis loops; inset shows the curve for pure BFO; and (b) variation of $M_{max}$, $M_r$ and $H_c$ for pure and doped BFO NPLs.

ceramics, resulting in a lossy unsaturated ferroelectric hysteresis loop [28]. As the substituting contents of Co and Ni increased, these unsaturated loops became broader with increase of their polarization. A linear increase in $P_r$ and $P_m$ with Co and Ni substitution in BFO can be seen from figure 4*b*, which shows an increase in uniform distribution of electric dipoles with substitution. Simultaneous increase in $P_r$ and $P_m$, confirms long-range ordering in doped BFO NPLs, indicating ferroelectric phase stability of the material despite of the substitution of Co and Ni, which is in coordination with the XRD analysis [29].

These NPLs were further selected to calculate the recoverable energy density ($W_R$) and energy loss density ($W_L$). The charge–discharge curves for pure and doped BFO are shown in figure 4*c*, while the variation of recoverable energy density and energy loss density are given in figure 4*d*. As the Co and Ni contents varied, the $W_R$ and $W_L$ were also increased. This increase may be related to the broader shape of the uni-polar loops obtained for doped BFO as shown in figure 4*c*. The highest values of $W_R$ and $W_L$ were obtained for the maximum Ni substitution in BFO. Higher values of $W_R$ can be attributed to larger breakdown strength (BDS) of these compounds [30]. The BDS of these ceramics depends upon the grain boundaries, grain size and the presence of other secondary phases. It has been reported that BDS is inversely proportional to the grain size of the material [31]:

$$BDS \propto \frac{1}{\sqrt{G}},\tag{3.1}$$

where $G$ is the grain size. Thus, a decreased grain size may result in higher values of BDS which could ultimately increase the $W_R$ [32].

*M–H* loops for pure and doped BFO NPLs are shown in figure 5*a*. For pure BFO, weak ferromagnetic character is clearly visible separately shown in the inset of figure 5*a*. This type of weak ordering has been reported previously for BFO NRs, which were quite different from the magnetic ordering of the bulk BFO [11,33]. Enhanced ferromagnetic behaviour with high coercive field was observed when BFO was co-doped with Co and Ni with maximum magnetization ($M_{Max}$) approaching saturation point nearly at 1 T magnetic field. A shift to strong ferromagnetic response was obtained with co-doping and the highest magnetization of approximately 1.7 emu g$^{-1}$ was observed for BFCNO-5. On the other hand, coercivity ($H_c$) of approximately 1700 Oe was observed for sample with only Co doping. $M_{Max}$, $H_c$ and retentivity ($M_r$) for all the synthesized samples are shown in figure 5*b*. This enhanced ferromagnetic behaviour with high $H_c$ and $M_r$ can be associated with formation of NPLs as they provide high surface area compared with other nanostructures. Previous reports suggest destabilization of NRs morphology with doping in BFO [11]. However, our approach of EG-mediated solvothermal process proved to be adequate for synthesizing functionalized nanostructures.

The origin of ferromagnetism in BFO nanostructures can be explained on the basis of the Goodenough–Kanamori rule [34]. In pure BFO, high spin 3*d* electron configuration of t$^3$e$^2$ at the Fe$^{3+}$ ions undergo Fe–O–Fe interactions, resulting in a lower symmetry state because of the octahedral rotations as shown in figure 6. These interactions result in virtual transfer of electrons, which have a tendency to hop from one site to other in between the overlapping orbitals through oxygen vacancies resulting in a super-exchange mechanism known for originating antiferromagnetic behaviour [35]. With the addition of Co and Ni, transfer of electrons can only happen if one of the cations in Fe–O–Co or Fe–O–Ni occupies low spin state with an empty e$^0$, resulting in ferromagnetic character. These

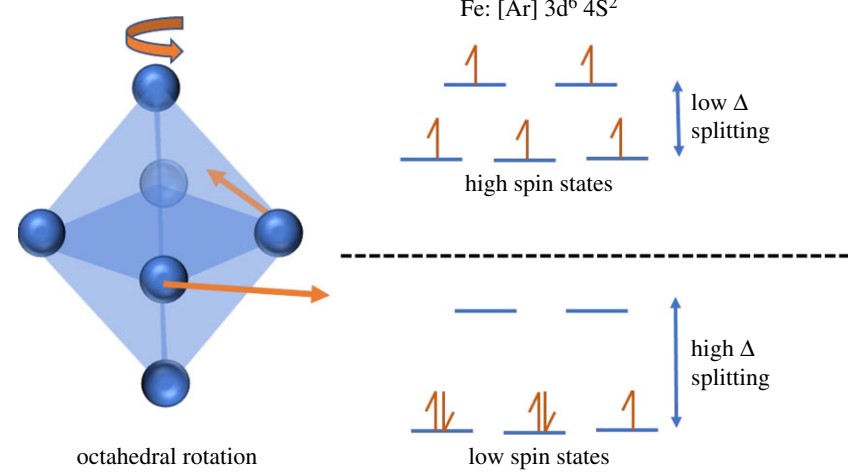

**Figure 6.** Octahedral rotation and formation of high and low spin states for Fe.

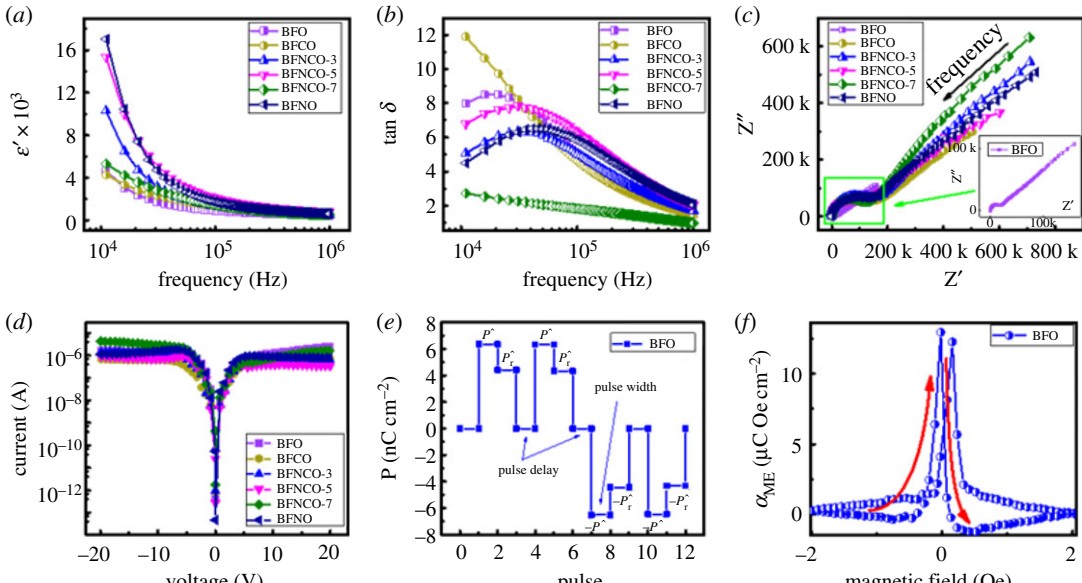

**Figure 7.** (a) Variation of $\varepsilon'$, (b) tan $\delta$, (c) Nyquist plot; inset shows the plot for pure BFO, (d) I-V plots for pure and doped BFO NPLs, (e) PUND sequence and (f) $\alpha_{Me}$ for pure BFO.

rules have been suggested by Goodenough [35] and Kanamori [36] in which they have shown a ferromagnetic Fe−O−Cr system. They have suggested that if $Fe^{3+}$ and $Cr^{3+}$ ions are introduced alternately in the B-site of $LaMnO_3$ perovskites, one can achieve a ferromagnetic material.

The frequency-dependent dielectric constant ($\varepsilon'$) and tangent loss (tan $\delta$) for Ni- and Co-doped $BiFeO_3$ NPLs are shown in figure 7a,b. A typical decreasing trend of dielectric dispersion with increasing frequency can be seen from figure 7a. Higher values of $\varepsilon'$ at lower frequencies correspond to space-charge polarization. It should be noted that the magnitude of $\varepsilon'$ observed for pure BFO NRs and doped BFO NPLs is much higher than the reported $\varepsilon'$ for BFO when synthesized in bulk. At higher frequencies, $\varepsilon'$ reduces and finally becomes saturated, indicating high stability at high frequencies. Similarly, tan $\delta$ shows a decreasing trend except at lower frequencies where a hump is visible. This hump has been characterized as dielectric relaxation mechanisms in much of the literature, previously reported [37]. High $\varepsilon'$ and low tan $\delta$ has been obtained for pure BFO NRs and doped BFO NPLs, which can be attributed to the low leakage current in these nanostructures [38]. Materials possessing high $\varepsilon'$ and low tan $\delta$ have shown remarkable interest for applications like on-chip capacitors, supercapacitors and other electrochemical devices [39,40]. Impedance spectroscopic results are shown in figure 7c in the form of Nyquist plots. At higher frequency, the intercept on the

Z′ axis shows the resistance of the sample, the diameter of the semicircle shows the charge transfer resistance and a constant phase element while a spike in the low-frequency region identifies the Warburg impedance. This type of Warburg impedance is usually obtained for energy storage materials and especially for NPLs [41]. This low-frequency vertical curve reaching close to 90° with high Z″ shows a good capacitive response for pure BFO as shown in the inset of figure 7c. As the Co and Ni doping increased, a sudden enhancement in diffusion resistance of ions occurred due to which high Z′ and Z″ values are obtained.

Figure 7d shows the leakage current measured in between ±20 V for all the samples. The leakage current for the doped BFO NPLs was found to be one order of magnitude lower than the pure BFO NRs while a good symmetry under positive and negative fields can be seen. This result resembles the impedance analysis for pure BFO in which a capacitive phenomenon is observed as the resistance was much lower than that of the doped BFO NPLs. Maximum value of leakage current is detected near ±6 V of the order of $10^{-6}$ A, after which the leakage current remains constant. Low values for leakage current and saturation after ±6 V demonstrates high stability of these NPLs even at high voltages. Leakage current measurements may contribute to calculate the switching charge densities ($Q_{SW}$) through pulse measurements [42]. For this reason, positive up and negative down (PUND) approach was adopted which proved quite feasible for determining the energy storage properties for capacitive materials [43]. Figure 7e shows the pulse measurements of pure BFO with certain pulse width of 10 ms and pulse delay time of 1000 ms. Polarization is measured after each pulse while the applied voltage for each pulse is 2 V. Total amount of polarization ($P^*$) was calculated at the first pulse (switching) which contains the contribution of both $Q_{SW}$ and leakage current. Second pulse (unswitching) measured the amount of polarization ($P^\wedge$) excluding the $Q_{SW}$. After these two pulses in positive direction, two negative pulses were applied, showing the polarization mechanism is the negative bias of the applied voltage. A very minute amount of difference was observed in the $P^*$, $P^\wedge$, $-P^*$ and $-P^\wedge$ values, respectively, which are tabulated along with the $Q_{SW}$ values in electronic supplementary material, table S2. The rest of the PUND measurements for doped BFO NPLs are shown in electronic supplementary material, figure S3. It can be seen that switching and unswitching values for all the samples are close to each other. However, the whole story is described by the trend of $Q_{SW}$ (electronic supplementary material, figure S3 and table S2), for the pure and doped BFO NPLs. $Q_{SW}$ value remained the same for all of the samples except for the last two compositions which showed a minute contribution of leakage current to the polarization reversal of the NPLs prepared.

Finally, ME coupling between the electrical and magnetic order parameters through ME coupling coefficient ($\alpha_{ME}$) for pure and doped BFO was measured. An AC magnetic field (H) was applied in order to determine variation in polarization across the BFO NPLs at a fixed frequency of 50 kHz as shown in figure 7f. The highest value of $\alpha_{ME}$ is found to be of 13 mV cm$^{-1}$ Oe$^{-1}$ at a bias field of −0.2 Oe followed by a sharp decrease with increase in H. Increase and decrease in $\alpha_{ME}$ is not symmetric with the application of H. This asymmetric behaviour of $\alpha_{ME}$ can be regarded to the asymmetricity in the piezo-response and magneto-strictive nature of the material [44]. $\alpha_{ME}$ for doped BFO NPLs is shown in electronic supplementary material, figure S4. Highest $\alpha_{ME}$ of 27 mV cm$^{-1}$ Oe$^{-1}$ was achieved for BFCNO-5 sample. Such type of $\alpha_{ME}$ versus H graphs has been reported previously [45]. These observations showed that a strong ME coupling was observed as these NPLs facilitate the RT polarization and magnetization phenomenon.

# 4. Conclusion

In this work, we synthesized hierarchical micro-sized flowers evolved by the self-assembly of highly crystalline Co- and Ni-doped BiFeO$_3$ NPLs, following a simple ethylene glycol-mediated solvothermal method. XRD analysis confirmed the rhombohedral perovskite structure with $R3m$ space group. Pure BiFeO$_3$ attained a scattered one-dimensional NR-like morphology of nearly 60 nm width and 200 nm lengths. Doping of BiFeO$_3$ with Co and Ni did not destabilize the morphology despite generating three-dimensional floral patterns of self-assembled two-dimensional NPLs. Unsaturated polarization loops with high coercivity were obtained for BiFeO$_3$ which confirmed the leakage behaviour of these rhombohedrally distorted cubic perovskites. A maximum polarization of 500 μC cm$^{-2}$ was obtained for BiFe$_{0.9}$Ni$_{0.1}$O$_3$. These hysteresis loops were then used to evaluate the recoverable energy and energy loss density. A maximum recoverable energy of 900 μJ cm$^{-3}$ was obtained. Enhanced ferromagnetic behaviour obtained for the synthesized BiFeO$_3$ was credited to the formation of high surface area NPLs. Impedance analysis showed a Warburg component for these NPLs. Switching/

non-switching plots show low leakage current while $\alpha_{ME}$ was also plotted in order to determine the ME coupling for the synthesized NPLs. $\alpha_{ME}$ for doped BFO NPLs have shown high values compared with pure $BiFeO_3$. Highest $\alpha_{ME}$ of 27 mV $cm^{-1}$ $Oe^{-1}$ was achieved for BFCNO-5 sample.

Data accessibility. Data are available at the Dryad Digital Repository: https://doi.org/10.5061/dryad.qjq2bvqcr [46].

Authors' contributions. S.K.A. contributed in project design, sample synthesis, experimental data and main article write-up. G.M.M. contributed in experimental data analysis and partial write-up. M.Sa. contributed in experimental data analysis. M.Su. contributed in providing experimental data and analysis. S.R. contributed in the provision of experimental data. S.N. contributed in experimental facilities and partial supervision. S.A. contributed in project design, overall supervision and proof reading.

Competing interests. The authors declare that they have no conflict of interest.

Funding. No funding was received for this work.

Acknowledgement. The authors are also thankful to Higher Education Commission (HEC), of Pakistan for providing partial experimental facilities under NRPU-2471, used in this work.

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
