## [Reviewer comments · Royal Society Open Science]

Review History

RSOS-200642.R0 (Original submission)

Review form: Reviewer 1

Is the manuscript scientifically sound in its present form?

Yes

Are the interpretations and conclusions justified by the results?

Yes

Is the language acceptable?

Yes

Do you have any ethical concerns with this paper?

No

Have you any concerns about statistical analyses in this paper?

No

Recommendation?

Accept with minor revision (please list in comments)

Comments to the Author(s)

The authors report the structure, magnetic and ferroelectric properties of BiFeO₃-based nanostructures. It is interesting in the field of multiferroics. The manuscript is well organized and clearly written. So, I recommend it to be published in Royal Society Open Science after some revisions.

1. The motivation of this manuscript should be strengthened by reviewing more previous results and cited, such as Nanoscale, 2020, 12, 477; etc.
2. In Fig. 4, the captions of (a)-(d) should be added.
3. In Fig. 6, in y axis, should μ_u be deleted? please check it carefully.
4. The error bar should be added in Fig. 7.

Review form: Reviewer 2**Is the manuscript scientifically sound in its present form?**

Yes

Are the interpretations and conclusions justified by the results?

No

Is the language acceptable?

Yes

Do you have any ethical concerns with this paper?

No

Have you any concerns about statistical analyses in this paper?

No

Recommendation?

Accept with minor revision (please list in comments)

Comments to the Author(s)

This manuscript reports the variation in multiferroic properties of hierarchical micro-sized flowers of highly crystalline Co and Ni doped BiFeO₃ nanoplates, synthesized using ethylene glycol mediated solvothermal method. This manuscript can be accepted for publication with the following minor clarifications.

1. Tables 1 and 2 as well as Figs 2 and 5 should be given as supplementary information.
2. In the introduction section the authors mention "Furthermore, doping of transition metal ions in BFO at A and B-sites have also shown strong variation in multiferroic properties." The following relevant reference should be cited: Dalton Transactions, 43 (2014) 7838-7846.
3. The authors should explain the reason for the choice of dopants. Why Co and Ni should have an effect on the magnetic and ferroelectric properties?
4. In results and discussion, the authors mention "a decreased grain size may result in higher values of BDS which could ultimately increase the WR." How do the authors confirm a decrease in the grain size with Ni doping?

5. "This enhanced ferromagnetic behavior with high H_c and M_r can be associated with formation of NPLs as they provide high surface area as compared to other nanostructures." BET surface area plots should be provided.

Decision letter (RSOS-200642.R0)

Dear Dr Atiq:

Title: Ethylene glycol assisted 3D floral evolution of BiFeO₃-based nanostructures with effective magneto-electric response
Manuscript ID: RSOS-200642

Thank you for submitting the above manuscript to Royal Society Open Science. On behalf of the Editors and the Royal Society of Chemistry, I am pleased to inform you that your manuscript will be accepted for publication in Royal Society Open Science subject to minor revision in accordance with the referee suggestions. Please find the reviewers' comments at the end of this email.

The reviewers and handling editors have recommended publication, but also suggest some minor revisions to your manuscript. Therefore, I invite you to respond to the comments and revise your manuscript.

Because the schedule for publication is very tight, it is a condition of publication that you submit the revised version of your manuscript before 21-Jun-2020. Please note that the revision deadline will expire at 00.00am on this date. If you do not think you will be able to meet this date please let me know immediately.

- 1) A text file of the manuscript (tex, txt, rtf, docx or doc), references, tables (including captions) and figure captions. Do not upload a PDF as your "Main Document".
- 2) A separate electronic file of each figure (EPS or print-quality PDF preferred (either format should be produced directly from original creation package), or original software format)
- 3) Included a 100 word media summary of your paper when requested at submission. Please ensure you have entered correct contact details (email, institution and telephone) in your user account

4) Included the raw data to support the claims made in your paper. You can either include your data as electronic supplementary material or upload to a repository and include the relevant doi within your manuscript

5) All supplementary materials accompanying an accepted article will be treated as in their final form. Note that the Royal Society will neither edit nor typeset supplementary material and it will be hosted as provided. Please ensure that the supplementary material includes the paper details where possible (authors, article title, journal name).

Kind regards,
Dr Laura Smith
Publishing Editor, Journals

On behalf of the Subject Editor Professor Anthony Stace and the Associate Editor Dr Dattatray Late.

RSC Subject Editor:
Comments to the Author:
(There are no comments.)

RSC Associate Editor:
Comments to the Author:
Authors reported micro-sized flowers with crystalline Co and Ni doped BiFeO₃ NPLs, following a simple ethylene glycol mediated solvothermal method. Authors need to add more supporting characterization data for X-RD. Here XPS analysis will give more accurate information about elemental analysis / bonding to focus on important findings of the manuscript.

Reviewer comments to Author:
Reviewer: 1

Comments to the Author(s)
The authors report the structure, magnetic and ferroelectric properties of BiFeO₃-based nanostructures. It is interesting in the field of multiferroics. The manuscript is well organized and clearly written. So, I recommend it to be published in Royal Society Open Science after some revisions.

1. The motivation of this manuscript should be strengthened by reviewing more previous results and cited, such as Nanoscale, 2020, 12, 477; etc.
2. In Fig. 4, the captions of (a)-(d) should be added.
3. In Fig. 6, in y axis, should μ be deleted? please check it carefully.
4. The error bar should be added in Fig. 7.

Reviewer: 2

Comments to the Author(s)

This manuscript reports the variation in multiferroic properties of hierarchical micro-sized flowers of highly crystalline Co and Ni doped BiFeO₃ nanoplates, synthesized using ethylene glycol mediated solvothermal method. This manuscript can be accepted for publication with the following minor clarifications.

1. Tables 1 and 2 as well as Figs 2 and 5 should be given as supplementary information.
2. In the introduction section the authors mention "Furthermore, doping of transition metal ions in BFO at A and B-sites have also shown strong variation in multiferroic properties." The following relevant reference should be cited: Dalton Transactions, 43 (2014) 7838-7846.
3. The authors should explain the reason for the choice of dopants. Why Co and Ni should have an effect on the magnetic and ferroelectric properties?
4. In results and discussion, the authors mention "a decreased grain size may result in higher values of BDS which could ultimately increase the WR." How do the authors confirm a decrease in the grain size with Ni doping?
5. "This enhanced ferromagnetic behavior with high H_c and M_r can be associated with formation of NPLs as they provide high surface area as compared to other nanostructures." BET surface area plots should be provided.

Author's Response to Decision Letter for (RSOS-200642.R0)

See Appendix A.

RSOS-200642.R1 (Revision)

Review form: Reviewer 1

Is the manuscript scientifically sound in its present form?

Yes

Are the interpretations and conclusions justified by the results?

Yes

Is the language acceptable?

Yes

Do you have any ethical concerns with this paper?

No

Have you any concerns about statistical analyses in this paper?

No

Recommendation?

Accept as is

Comments to the Author(s)

It is acceptable.

Review form: Reviewer 2

Is the manuscript scientifically sound in its present form?

Yes

Are the interpretations and conclusions justified by the results?

Yes

Is the language acceptable?

Yes

Do you have any ethical concerns with this paper?

No

Have you any concerns about statistical analyses in this paper?

No

Recommendation?

Accept as is

Comments to the Author(s)

The authors have tried to incorporate most of the recommendations suggested by the reviewers. Though BET surface area could not be reported due to lack of access, the quality of the revised manuscript has improved and can be accepted for publication in Royal Society Open Science.

Decision letter (RSOS-200642.R1)

Dear Dr Atiq:

Title: Ethylene glycol assisted 3D floral evolution of BiFeO₃-based nanostructures with effective magneto-electric response

Manuscript ID: RSOS-200642.R1

It is a pleasure to accept your manuscript in its current form for publication in Royal Society Open Science. The chemistry content of Royal Society Open Science is published in collaboration with the Royal Society of Chemistry.

On behalf of the Subject Editor Professor Anthony Stace and the Associate Editor Dr Dattatray Late.

RSC Associate Editor:
Comments to the Author:
Accept as is

RSC Associate Editor:
Comments to the Author:
missing all supplementary files

Reviewer(s)' Comments to Author:
Reviewer: 1

Comments to the Author(s)
It is acceptable.

Reviewer: 2

Comments to the Author(s)
The authors have tried to incorporate most of the recommendations suggested by the reviewers. Though BET surface area could not be reported due to lack of access, the quality of the revised manuscript has improved and can be accepted for publication in Royal Society Open Science.

Appendix A

RESUBMISSION OF MANUSCRIPT REF # RSOS-200642

Dr. Laura Smith

Publishing Editor,

Dear Madam,

Many thanks for sending us your response to our submission **RSOS-200642**. We have read the reviews and would like to thank the reviewers for their valuable remarks and comments to improve the quality of the research presented in this manuscript. So, a revised form of this manuscript is now being submitted while the additions were highlighted as **yellow**.

Response to Reviewer's Comments

Reviewer # 1

The authors report the structure, magnetic and ferroelectric properties of BiFeO₃-based nanostructures. It is interesting in the field of multiferroics. The manuscript is well organized and clearly written. So, I recommend it to be published in Royal Society Open Science after some revisions.

Comment # 1

The motivation of this manuscript should be strengthened by reviewing more previous results and cited, such as Nanoscale, 2020, 12, 477; etc.

Our response:

We thank the reviewer for appreciating and encouraging our efforts. We have now added the references as suggested by the reviewer.

Comment # 2

In Fig. 4, the captions of (a)-(d) should be added.

Our response:

The captions have been added.

Comment # 3

In Fig. 6, in y axis, should μ_u be deleted? please check it carefully.

Our response:

Yes, μ_u has been removed from the Y-axis.

Comment # 4

The error bar should be added in Fig. 7.

Our response:

The error bar has been added in Fig. 7 which is now Fig. 5

Reviewer # 2

This manuscript reports the variation in multiferroic properties of hierarchical micro-sized flowers of highly crystalline Co and Ni doped BiFeO₃ nanoplates, synthesized using ethylene glycol mediated solvothermal method. This manuscript can be accepted for publication with the following minor clarifications.

Comment # 1

Tables 1 and 2 as well as Figs 2 and 5 should be given as supplementary information.

Our response:

The Tables and Figs have now been adjusted as the supplementary information.

Comment # 2

In the introduction section the authors mention “Furthermore, doping of transition metal ions in BFO at A and B-sites have also shown strong variation in multiferroic properties.” The following relevant reference should be cited: Dalton Transactions, 43 (2014) 7838-7846.

Our response:

The reference has been added.

Comment # 3

The authors should explain the reason for the choice of dopants. Why Co and Ni should have an effect on the magnetic and ferroelectric properties?

Our response:

Co and Ni are considered as fundamental magnetic elements that are used to increase the magnetic properties of the materials under investigation. On the other hand, ferroelectric properties have been found to increase by increasing the doping contents. So, this doping family can be the ideal case for both the properties and our results suggests the above assumptions to be true.

Comment # 4

In results and discussion, the authors mention “a decreased grain size may result in higher values of BDS which could ultimately increase the WR.” How do the authors confirm a decrease in the grain size with Ni doping?

Our response:

From Fig. 2 it can be seen that the average size of nanoplates decreases as we increase the doping contents from Fig. 2b to 2f. We have used FESEM technique to determine the sizes of these plates. The sizes were cross checked using a Java based ImageJ software.

Comment # 5

“This enhanced ferromagnetic behavior with high H_c and M_r can be associated with formation of NPLs as they provide high surface area as compared to other nanostructures.” BET surface area plots should be provided.

Our response:

We are sorry we don't have access to BET surface area technique but generally, we have claimed this statement as these nanostructures are in the form of nanorods and nanoplates which usually have high relative surface areas.